# Multi-instance Interactive Segmentation with Self-Supervised Transformer

## Abstract

The rise of Vision Transformers (ViT) combined with better self-supervised learning pre-tasks has taken representation learning to the next level, beating supervised results on ImageNet. In particular, self-attention mechanism of ViT allows to easily visualize semantic information learned by the network. Following revealing of attention maps of DINO, many tried to leverage its representations for unsupervised segmentation. Despite very promising results for basic images with a single clear object in a simple background, representation of ViT are not able to segment images, with several classes and object instance, in an unsupervised fashion yet. In this paper, we propose SALT: Semi-supervised Segmentation with Self-supervised Attention Layers in Transformers, an interactive algorithm for multi-class/multi-instance segmentation. We follow previous works path and take it a step further by discriminating between different objects, using sparse human help to select said objects. We show that remarkable results are achieved with very sparse labels. Different pre-tasks are compared, and we show that self-supervised ones are more robust for panoptic segmentation, and overall achieve very similar performance. Evaluation is carried out on Pascal VOC 2007 and COCO-panoptic. Performance is evaluated for extreme conditions such as very noisy, and sparse interactions going to as little as one interaction per class.

## 1 Introduction

The last ten years have seen the rise of computer vision tasks such as localization and segmentation. As a result, technologies such as autonomous driving or robotics have met great success at the expense of annotating huge enough datasets. Indeed, state-of-the-art approaches are all based on training a neural network in a supervised fashion (Strudel et al. 2021, Xie et al. 2021). Although this might work well in areas where there are enough resources to label million of images, there are others where there are almost no labels but data is already available in large quantities. For instance, in fields such as astronomy, sometimes one is limited by the amount of available ground truth labels (Pasquet et al., 2019). On other ones like medical imaging, data needs to be labeled by professionals, which is very expensive. Therefore, leveraging unlabeled data is a necessity in many computer vision tasks.

Numerous attempts exist in the literature to solve this problem, such as semi-supervised learning (Kipf & Welling, 2017), weakly supervised learning (Strudel et al., 2022), and active learning (Aghdam et al., 2019). These methods can achieve some improvement, but still need slight supervision. More recently, self-supervised pre-tasks have leveraged representation power of Vision Transformer (ViT: Dosovitskiy et al. 2021) in a similar fashion to what has been done in NLP. Indeed, an image can be seen as a sequence of $p \times p$ patches. Transformers have recently outperformed convolutional neural networks, and results with self supervised pre-task DINO (Caron et al., 2021) have shown impressive salient regions in the attention maps from the class token in the last ViT layer. This has led authors to test the unsupervised foreground detection capabilities of such representations (Wang et al. 2022; Amir et al. 2021). Authors tried clever ways to cluster these feature representations to split foreground and background regions. Melas-Kyriazi et al. (2022) went one step ahead and tried to do this for more than one foreground object. However, these applications are limited to simple images with a clear background and very few salient objects.

We believe that the representation power of self-supervised ViTs can be pushed much further with very sparse human interactions. Here, we try to discriminate between different objects, using sparse human help to select said objects. Our goal is twofold, we want to assess the representation power of self-supervised ViTs, and at the same time create an interactive segmentation algorithm that is not powered by a supervised learning algorithm, but still harnesses information from a dataset. Indeed, if one has a huge unlabeled dataset, then self-supervised learning could be first used to derive meaningful representations, and our algorithm could help label images in just a few seconds.

In this paper, we propose SALT: Semi-supervised Segmentation with Self-supervised Attention Layers in Transformers. A graph based semi-supervised approach that harnesses the representation power of self-supervised ViTs to create segmentation masks from sparse human interactions. We test segmentation performance on Pascal VOC 2007 (Everingham et al., 2010), and a modified version of COCO-panoptic (Lin et al., 2015b) that contains only big *things*, hereafter *COCO big things*. However, because of the nature of interactive segmentation algorithms themselves, we will not be comparing to other algorithms. Indeed, the major obstacle is that algorithms usually take different forms of inputs, and can also be iterative. To the best of our knowledge, there are no interactive segmentation methods that share the same input as ours. For each dataset, we will create an interaction dataset with human inputs, and we will craft elaborated evaluation protocols that show what can be realistically expected from our algorithm, as well as its limitations.

We study the performance of our algorithm in extreme conditions, which can be interesting for some unsupervised tasks where we have very sparse information of the position of an object. To the best of our knowledge, this is the first unsupervised interactive segmentation algorithm that is able to handle many panoptic classes simultaneously, while achieving pleasing results. Although results are still far below the performance of supervised state-of-the-art algorithms, this work shows the potential of future ViTs for zero-shot interactive segmentation, and eventually unsupervised segmentation.

The paper is organized as follows. In section 2, we present the prerequisites. In section 3, we explain our method. In section 4, we present the modifications made to COCO-panoptic, and how we gathered interactions. In section 5, we compare different pre-trained ViTs, and evaluate the robustness of our method to noise and interactions sparsity up to one patch per class. Finally, in section 6 we present our conclusions.

## 2  RELATED WORK

**Vision Transformer.** Transformer architecture (Vaswani et al., 2017) has become the default architecture for natural language processing (NLP) since it was first introduced five years ago. It was not until very recently that computer vision started transitioning from Convolutional Neural Networks (LeCun et al., 1998) to Transformers. Pioneer works tried to implement the self-attention mechanism within CNNs (Hu et al. 2019; Ramachandran et al. 2019; Zhao et al. 2020). Dosovitskiy et al. (2021) ultimately released the Vision Transformer, using $16 \times 16$ patches as tokens, and almost the same encoder as in the original Trasformer. Since they first appeared, a lot of strategies have been developed to train ViTs more effectively (Beyer et al. 2022, Touvron et al. 2021, 2022), as well as variants (Liu et al., 2021). Besides, many recent work have shown that ViTs trained in a self-supervised fashion (Caron et al. 2021, Bao et al. 2021, Assran et al. 2022) outperform their supervised counterpart. Self-supervised ViT attention maps have also shown high semantic comprehension.

**Self-supervised learning.** In recent years, different clever pre-task have been developed to exploit unlabeled data and pre-train models in a self-supervised fashion. Pioneering works designed ingenious pretext tasks to exploit internal structures of data, such as patch ordering prediction (Noroozi & Favaro, 2017), recovering colors from grayscale images (Zhang et al., 2016), image rotation prediction (Gidaris et al., 2018), etc. Nowadays, most approaches fall into one of two categoriers: generative or discriminative. Generative methods are usually based on masked image encoding (He et al. 2021, Bao et al. 2021). Contrastive learning usually uses siamese networks to discriminate between two views of an image (Chen et al. 2020, He et al. 2020, Grill et al. 2020, Caron et al. 2021, Assran et al. 2022).

**Unsupervised Segmentation.** Earlier methods mostly used color/background constraints (Cheng et al. 2014, Wei et al. 2012). Recently, methods based on extracting features using a self-supervised Transformer (Dosovitskiy et al., 2021) based on DINO (Caron et al., 2021) significantly improved

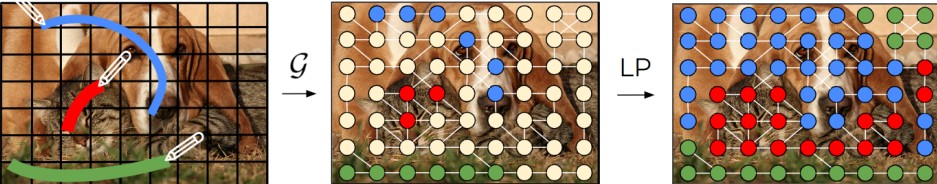

Figure 1: Sketch of how the graph is built, and how the human interactions are used. Here, each node corresponds to a ViT patch, and each color is a different class. Then, the graph with sparse labels goes through label propagation (LP) that predicts unlabeled nodes.

over state-of-the-art for unsupervised object discovery and segmentation (LOST: Siméoni et al. 2021; TokenCut: Wang et al. 2022). These methods use ViT features to create a graph $\mathcal{G}(V, E)$ where nodes are patches, and edges are defined according to the similarity of the features. However, these methods cannot work in images that have more than one salient object. More recently, Melas-Kyriazi et al. (2022) generalized this approach to more than one object. However, there is much work to do before these methods can properly segment a complex multi-instance image.

**Interactive Segmentation.** Interactive segmentation methods attempt to use external information to extract foreground objects in a complex environment whose background cannot be trivially subtracted. This information can be passed in many different ways, mainly (i) extreme points , (ii) bounding boxes, (iii) scribbles, (iv) positive/negative points. Earlier methods used color statistics (Magic Wand: Gutiérrez & Vexo 2003, Bayes matting: Yung-Yu Chuang et al. 2001), edge contrast (Intelligent Scissors: Mortensen & Barrett 1995), Maxflow optimization (GrabCut: Rother et al. 2004, Saliency Cuts: Fu et al. 2008, Bai & Wu 2014), optimal transport (Rabin & Papadakis, 2015). These methods are all limited by the information contained on the image, and as a result, they often need many interactions to achieve a satisfactory result. Deep learning based methods use point-based interactions such as extreme points (Xu et al. 2016, Papadopoulos et al. 2017, Zhang et al. 2020), positive/negative points (Mahadevan et al. 2018, Hao et al. 2021, Zhang et al. 2022), and as little as two points (UCP-Net: Dupont et al. 2021). However, there still does not exist a multi-instance interactive segmentation algorithm that leverages information contained in a dataset.

## 3 APPROACH

The general pipeline of our approach is summarized in Figure 1. A user draws scribbles to select different objects in an image. Then, the patch feature representation are extracted from the pretrained ViT and a graph $\mathcal{G}$ is built. Finally, the user labels are used along with $\mathcal{G}$ as input for the Label Propagation (LP: Zhou et al. 2004) algorithm, which predicts the remaining labels.

**Label propagation.** Unlike unsupervised methods, we want to explore the limits of self-supervised representations in more complex scenarios. By using sparse human interactions, we can therefore use more complex images with lots of objects/instances. As a result, we are in a semi-supervised learning scenario. Label propagation (Zhu & Ghahramani 2002, Zhou et al. 2004) are a kind of graph based semi-supervised learning algorithms. The underlying idea is that since we do not have labels for each node, known ones ares used to propagate information through the graph and label remaining nodes. Let $X = (x_1, x_2, \ldots, x_n)$ be our dataset with $n$ examples. In semi-supervised learning, we only have a fraction $X_l = (x_1, \ldots, x_l)$, for which labels $Y_l = (y_i)_{i \in \{1, l\}}$ are provided, and a fraction $X_u = (x_{l+1}, \ldots, x_{l+u})$ for which we do not know the labels. The algorithm computes the normalized graph Laplacian matrix and predicts labels $\hat{Y} = (\hat{Y}_l, \hat{Y}_u)$ iteratively. The algorithm pseudo-code is presented in Algorithm 1.

The cost function associated with this problem is

$$C(\hat{Y}) = \frac{1}{2} \left( \sum_{i,j} \boldsymbol{A}_{ij} \left\| \frac{\hat{y}_i}{\sqrt{\boldsymbol{D}_{ii}}} - \frac{\hat{y}_j}{\sqrt{\boldsymbol{D}_{jj}}} \right\|^2 + \mu \sum_{i=1}^{n} \|\hat{y}_i - y_i\|^2 \right) \tag{1}$$

where $\boldsymbol{A}$ is the affinity or adjacency matrix, $\boldsymbol{D}$ is the diagonal degree matrix, and $\mu > 0$ is the regularization parameter. The first term is the smoothness assumption, which ensures consistency

---

**Algorithm 1** Label spreading (Zhou et al., 2004)

---

**Require:** $\alpha \in [0, 1[$
    Compute affinity matrix $\boldsymbol{A}$, $A_{ii} \leftarrow 0$
    Compute diagonal degree matrix $\boldsymbol{D}$: $D_{ii} \leftarrow \sum_j A_{ij}$
    Compute normalized graph Laplacian $\mathcal{L} \leftarrow \boldsymbol{D}^{-1/2}\boldsymbol{A}\boldsymbol{D}^{-1/2}$
    Initialize $\hat{Y}^{(0)} \leftarrow (y_1, \dots, y_l, 0, 0, \dots, 0)$
    **for** $t$ in range $n_{iter}$ **do**
        $\hat{Y}^{(t+1)} \leftarrow \alpha\mathcal{L}\hat{Y}^{(t)} + (1-\alpha)\hat{Y}^{(0)}$
        **if** convergence **then**
            **break**
        **end if**
    **end for**
    **return** Label $x_i$ as a label $\operatorname{argmax} \hat{y}_i^{(\infty)}$

---

with the geometry of the data. The second term is the fitting constraint, which penalizes rapid changes in $\hat{Y}$ between points that are close. Zhu & Ghahramani (2002) forces $\hat{Y}_l = Y_l$. However, if there is noise on the labels, the algorithm should be able to re-label them. The same kind of cost criterion is obtained for spectral clustering for the relaxed NP-hard problem of minimizing the normalized cut of $\mathcal{G}$.

**Feature representation.** Our goal is to leverage self-supervised learning methods representation power. It has been shown that Vision Transformers pretrained in such ways create powerful representations that can be used to segment salient objects in images (e.g. Siméoni et al. 2021; Amir et al. 2021; Wang et al. 2022; Melas-Kyriazi et al. 2022). These methods are all based on some clustering algorithm that will separate the salient object from the background. All of these methods are based on DINO (Caron et al., 2021), and conclude through an ablation study that the best feature representation are the *keys* from the last ViT-B layer. Here, we will use Masked Siamese Networks (MSN: Assran et al. 2022), a very similar pre-tasks to DINO. We will also use *keys* from the last ViT-B/16 layer, although we found no significant improvement compared to other representations in the last layers (see Appendix A).

**Graph construction.** Following Siméoni et al. (2021), we use ViT features to build a graph $\mathcal{G}(V, E)$ where each node $V$ represents a patch. We consider three different approaches. (i) We define edges $E_{i,j}$ between two nodes $V_i, V_j$ as (Wang et al., 2022)

$$E_{ij} = \begin{cases} 1 & \text{if } S(i, j) \geq \tau \\ \epsilon & \text{otherwise} \end{cases} \tag{2}$$

where $S$ is a similarity score based on the cosine similarity of the feature vectors of the two patches, $\tau \in ]0, 1[$ is a hyper-parameter, and $\epsilon$ is very small to ensure the graph is fully connected. (ii) A Gaussian kernel such that the adjacency matrix is

$$\boldsymbol{A}_{i,j} = exp\left(-\frac{\|x_i - x_j\|^2}{2\sigma^2}\right) \tag{3}$$

(iii) A K-nearest neighbors graph. We will refer to these graphs as *similarity*, *RBF*, and *KNN* respectively.

We also tried to add low level features such as color information to the graph, as in Melas-Kyriazi et al. (2022)

$$\boldsymbol{A} = \boldsymbol{A}_{feat} + \lambda\boldsymbol{A}_{color}$$

where $\boldsymbol{A}_{feat}$ is the adjacency matrix for the graph defined using ViT features, $\lambda$ is a hyperparameter, and $\boldsymbol{A}_{color}$ is a KNN graph defined using color information. However, it did not show any improvement in our results. Experiments will be carried out using a RBF graph with no color information, and an ablation study is available in Appendix B.

**Higher resolution.** We propose to effectively double the patch resolution by computing the linear interpolation between patch representations. Amir et al. (2021) proposed to take overlapping patches. Indeed, by passing the image through a single convolution layer, with stride that equals

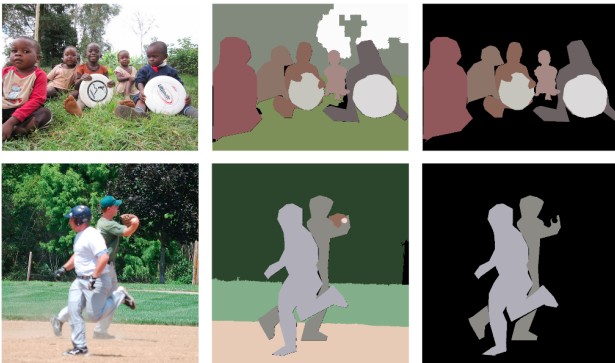

Figure 2: (left) Two COCO-panoptic images from the validation set. (middle) COCO-panoptic labels. (right) COCO-panoptic Big Things labels: stuff and small things are merged with the background.

half the patch size, we achieve overlapping patches, and a patch resolution half the original size. However, this comes at the expense of GPU memory, and only achieves slightly better results (mIoU 81.3 instead of 81.0 on MSN for VOC07). It is also possible to use ViT-B/8 architecture instead of ViT-B/16 with resolution increase. However, this usually achieved worst results. For instance, we achieved mIoU of 76.8 on DINO-B/8, and 80.8 on DINO-B/16 for VOC07. Therefore, we did not use these techniques and used our linear interpolation instead.

Even if this allows us to reduce the patch size by half, we still need a way to achieve pixel-level segmentation. We cannot decrease to a resolution below $p = 8$ as it would become computationally too expensive. Krähenbühl & Koltun (2012) developed a method to label every pixel in the image with one of several predetermined object categories. Then the problem is solved as a maximum a posteriori (MAP) inference in a conditional random field (CRF) defined over pixels or image patches. Here, we use our patch-level mask as input and mask $20\%$ patches with an uniform random distribution.

## 4 EVALUATION

### 4.1 DATASETS

We will carry out evaluations on Pascal VOC 2007 (Everingham et al., 2010) train-val set, and COCO-panoptic (Lin et al., 2015a) validation dataset. COCO usually has many complex scenes with multiple objects. There are two main categories of objects: *things* (e.g. persons, indoor objects, etc.) and *stuff* (e.g. grass, sky, etc.). Manually labeling all these objects would be very time expensive, and would bring little additional information regarding the performance of our model. Therefore, we keep only *things*. Besides, we remove objects with less than $0.5\%$ of total pixels, as small objects are also time expensive to label. We remove images for which there are no remaining objects. These modifications are performed on the validation set, and we call our subset *COCO-panoptic big things*. Figure 2 shows an example of the COCO-panoptic dataset before, and after our modifications.

### 4.2 INTERACTIONS

Properly evaluating an interactive segmentation algorithm is a very subtle task. Depending on the kind of interaction, they can be simulated fairly easy. For instance, simulating extreme points (Papadopoulos et al., 2017), or sparse point interactions (Dupont et al., 2021) can be done with very little assumptions. However, simulating more complex interactions such as scribbles is an active field of research (Jiang et al., 2016). Besides, the amount of interactions is not the only important parameter. Indeed, one should also consider the time spent to label an image, for not all interactions take the same time. For instance, a straight line can be done faster than aiming for a particular point. However, time cannot be easily compared between annotators, and we believe using it in a metric would not bring any relevant information. Therefore, here we mostly focus on the interactions alone.

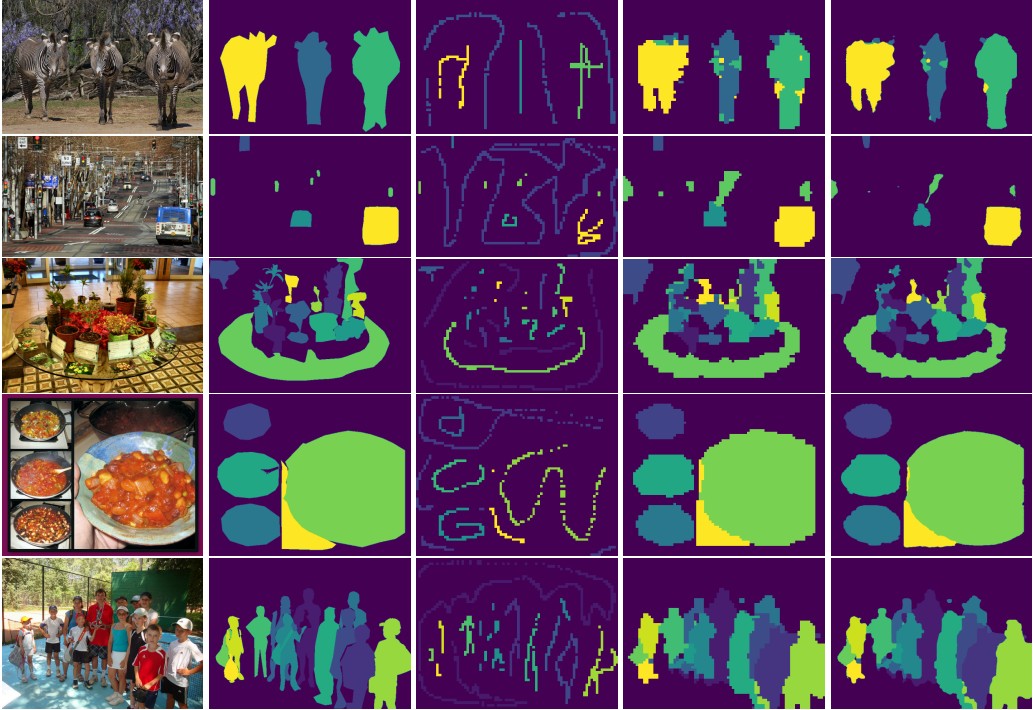

Figure 3: COCO-Panoptic results. From left to right, image, ground truth, handmade interactions, predicted labels (patch scale), and predicted labels (pixel scale).

To the best of our knowledge, there is no effective method that properly simulates human interactions. Therefore, we decided to manually annotate these datasets, for there is no better way to evaluate human interactions than actually asking humans to do them. We tried to give simple instructions to annotators, so that annotations remained truthful. We developed a custom software to annotate images according to their real labels. We asked annotators to spend around fifteen seconds per image on COCO, and less than five on VOC07. The instructions read as follows

- Do not spend more than a few seconds per image.
- If the scene is complex, try to draw scribbles in regions of interest that contrast with objects.
- You do not have to label all the different components of an instance (for semantic labels only), but you can if you want as long as it does not take too much time.
- You can draw scribbles but you can also click once if you feel the area is too small.
- You can go over the edges if the area is too small, but try to keep within the area as much as possible.

Interactions for VOC07 were made by one person only (4 seconds per image on average). Interactions for COCO-panoptic where made by 15 different people (16 seconds per image on average), and amount to 1423 out of 4823 images.

## 5 EXPERIMENTS

In this section we present different experiments we performed to evaluate our method. First, we compare different pre-trainings and see how they perform. Then, we study the robustness of our method against noise, as well as against using a fraction of interactions. Finally, we test our algorithm under extreme conditions using only one patch per class. We use the `timm` (Wightman, 2019) library for ViT architecture, and `Scikit-learn` implementation of the Label spreading algorithm (Zhou et al., 2004).

## 5.1 Impact of the pre-training

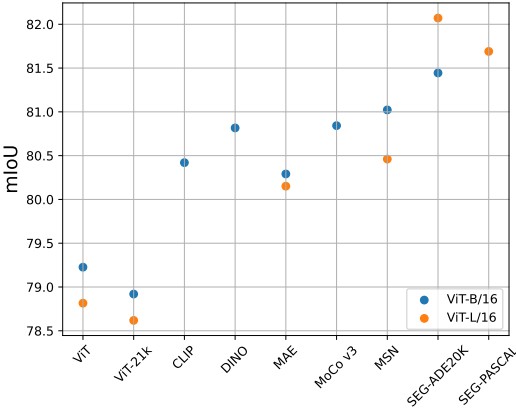

Figure 4: Pascal VOC 2007 analysis over all pre-trainings.

State-of-the-art methods in unsupervised segmentation (Siméoni et al. 2021; Amir et al. 2021) showed through ablation studies that using *keys* from the last ViT-B/16 layer as patch representations yielded best results for DINO pre-trainings. We conducted an ablation study of our own in Appendix A, and found that the deeper layer, the better result. However, the gap between *keys* and *queries* or *values* is very small and does not suggest that *keys* are better overall. Still, we will conduct analysis using *keys* from the last ViT layer, for both base and large architecture.

We compare results for different kind of pre-trainings on ViT-B/16 and ViT-L/16. We use vanilla ViT pretrained on ImageNet-1k and ImageNet-21k (Dosovitskiy et al., 2021); CLIP (Radford et al., 2021), a text-to-image pre-training that jointly trains an image encoder, and a text encoder to predict the correct image/text pairs; Self supervised methods pre-trained on ImageNet-1k: DINO (Caron et al., 2021), MAE (He et al., 2021), MoCo-v3 (He et al., 2020), and MSN (Assran et al., 2022); and ViT encoders trained with Segmenter (Strudel et al., 2021), a supervised method for image segmentation, on ADE20k and Pascal Context.

Figure 4 shows results for Pascal VOC 2007. We can see that there is a clear gap in performance between vanilla ViTs and self-supervised methods. Also, Segmenter ViT beats self-supervised methods for this task. Figure 5 shows results for COCO-panoptic using panoptic, and semantic labels[1]. We can see that the gap between vanilla pre-trainings and self-supervised ones increases for panoptic labels. Moreover, Segmenter ViTs are on par with self-supervised ones for semantic labels, but are beaten for panoptic ones. Therefore, self-supervised pre-trainings seem to be better suited at panoptic segmentation.

## 5.2 Noise robustness

An important feature of our algorithm is its capacity to perform well even if there are incorrectly labeled patches within the interaction. Our handmade interactions already contain wrongly labeled patches, as seen in Appendix C. Our goal here is to analyse how much we can deteriorate the interactions, while still achieving good results. To do so, we add noise to each handmade interaction independently. The background is usually labeled without problems. Therefore, we do not add noise to it.

For each mask in a label, we define a density map $P$ for each patch $x$

$$P(x) = \begin{cases} \exp\left(-\dfrac{d_{\mathcal{S}}(x)^2}{2\sigma^2}\right) & \text{if } x \notin \mathcal{S} \\ 0 & \text{otherwise} \end{cases} \tag{4}$$

---

[1]Semantic labels are created from panoptic ones. We do not use the existing COCO labels for semantic segmentation, which are slightly different.

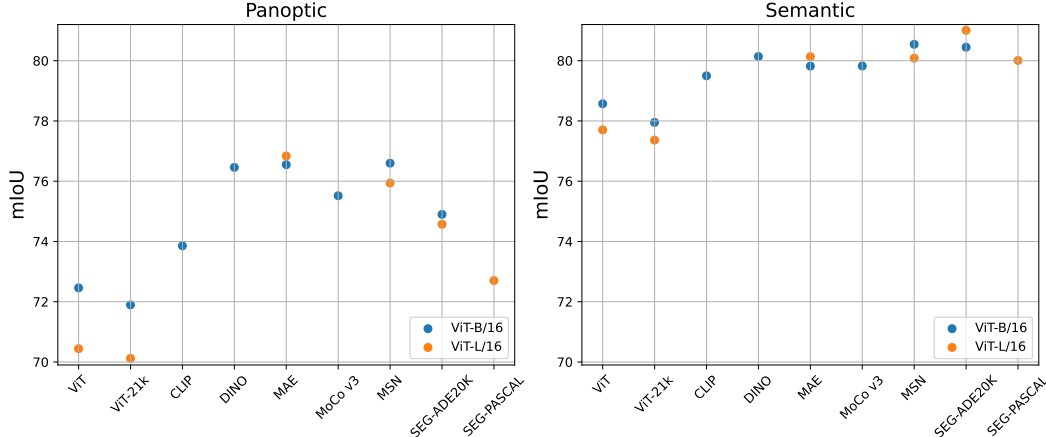

Figure 5: COCO-panoptic v. semantic analysis over all pre-trainings.

where $\mathcal{S}$ is the mask, $\sigma$ is a parameter we set to 4 to keep the simulation realistic, and $d_{\mathcal{S}}(x)$ is the shortest distance between $x$ and $S$. We sample *noisy* patches from this distribution.

Results for COCO-panoptic are shown in Figure 6. For each interaction, a percentage $p \in [0, 0.5]$ of unlabelled patches are added to the interaction uniformly at random. This percentage corresponds to a fraction of the number of labeled patches per object. For instance if noise = 0.1, then for each interaction, unlabeled patches accounting for $10\%$ of the total number of labeled patches are added to the interaction. Overall, the algorithm does not seem to be robust to extra noise. However, we usually do not have more than $5\%$ extra noise per category. Thus, our algorithm is robust enough for interactive segmentation.

## 5.3 SPARSE LABELS ROBUSTNESS

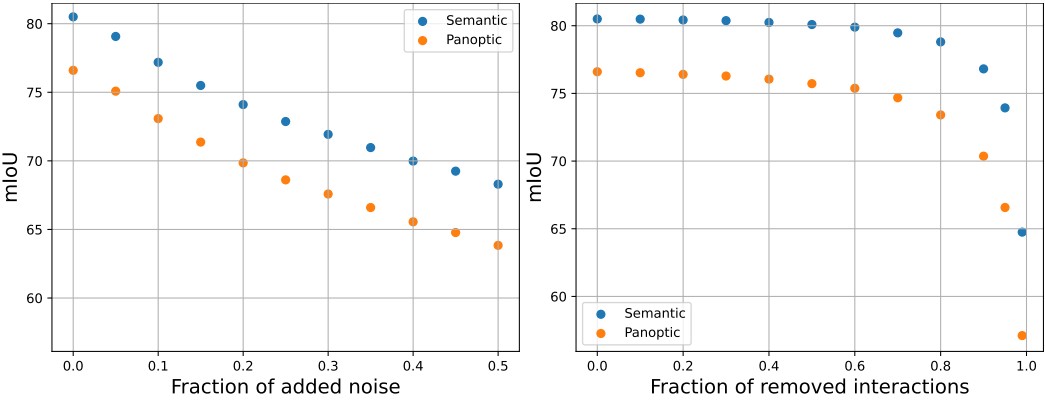

Figure 6: (right) COCO noise robustness analysis. (left) Analysis of robustness against using a fraction of the handmade labels. The masked $\%$ corresponds to the fraction of labeled patches that is removed in an image for each individual instance.

We test our model robustness against using a fraction of the handmade labels. For each instance, we remove uniformly at random $p\%$ patches from the interactions, always leaving at least one patch per instance. Figure 6 shows results for $p \in [0, 0.99]$. The algorithm does not suffer a drop in performance until we remove more than $60\%$ of patches per instance, and performs remarkably well until $80\%$. The drop is more important for panoptic labels than for semantic ones. Indeed, using more labels forces the algorithm to properly separate two instances of a same class. If one leaves only a few labels per instance, the algorithm could be inclined to discriminate between the different

parts of the objects instead (e.g. separating the head from the body). Hence, the algorithm needs more labels to achieve good results in a panoptic configuration.

### 5.4 ONE PATCH TO RULE THEM ALL

We simulate interactions uniformly at random using one labeled patch per class, including the background. This is the lowest amount of interactions that we can give to our algorithm, and as a result, the most extreme conditions. Using only one patch per class means that if we have an algorithm that finds seeds for different objects, this is the highest performance we could expect for said unsupervised algorithm. Figure 7 shows results for a few images. We can see that using only one patch in an image gives many possible correct labels. For instance, on the second image from Figure 7, there are many objects but only three are labeled. However, the algorithm selects them all and spreads them accross the three labels. Therefore, the algorithm would need accurate labels for all objects in an image to be properly evaluated. We performed this experiment five times and achieved an mIou of $61.7 \pm 0.5$ for VOC07, and $50.3 \pm 0.8$, $56.7 \pm 0.7$ respectively for COCO panoptic and semantic. Thus, the algorithm performs remarkably well in extreme conditions.

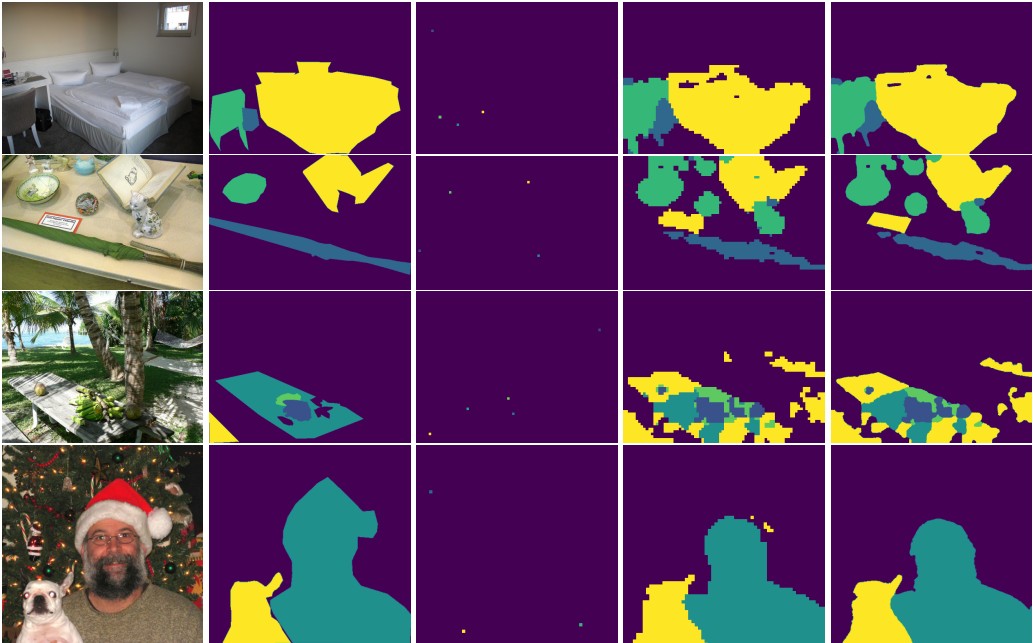

Figure 7: Same as Figure 3 with one patch per class.

## 6 CONCLUSION

In this paper, we have introduced a method, SALT, for multi-instance interactive segmentation leveraging self-supervised learning, and Label Propagation. This work goes along the path of previous works that show the power of graph theory. We have shown that self-supervised representations can accurately distinguish, with relatively sparse human interactions, between different classes and instances within a class. We achieved this by evaluating performance on two datasets: VOC07 and a modified version of COCO panoptic that only includes big things. Our main results in this work include (i) Self-supervised learning pre-trainings are more robust for panoptic segmentation (ii) the algorithm is not very robust to noise, but is good enough for the expected amounts of noise (iii) the algorithm can achieve very good performance with very limited interactions (iv) the algorithm performs remarkably well in the most extreme conditions using only one patch per label. All in all, we believe that with better pre-tasks to come, this technique will help label datasets much faster, and this method could also be generalized for unsupervised segmentation if coupled with an object detection algorithm.

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

# A  BACKBONE ANALYSIS

State-of-the-art methods in unsupervised segmentation (Siméoni et al. 2021; Amir et al. 2021) showed through ablation studies that using *keys* from the last ViT-B/16 layer as patch representations yielded best results for DINO pretrainings. Here we will see if this is true for our task, and see if it is also true for other pretrainings. Our analysis are all conducted on Pascal VOC 2007 (Everingham et al., 2010).

Table 1: mIoU analysis of the different ViT-B/16 backbones for each feature of each layer. For each layer $\mathcal{L} \in [0, 11]$, each row corresponds to *keys*, *queries*, and *values*. Four supervised methods (vanilla ViT and ViT-21k pretrained on ImageNet-1k, and -21k respectively: Dosovitskiy et al. 2021; CLIP, pretrained on OpenAI custom dataset: Radford et al. 2021; Segmenter (Strudel et al., 2021) trained on ADE20K). Four self-supervised methods pretrained on ImageNet-1k (DINO: Caron et al. 2021; MAE: He et al. 2021; MoCo-v3: He et al. 2020; MSN: Assran et al. 2022). For each column, Green corresponds to best results, blue to the following five best, and red to the three worst.

| Layer | Supervised | | | | Self-supervised | | | |
|---|---|---|---|---|---|---|---|---|
| $(k, q, v)$ | ViT | ViT-21k | CLIP | Segmenter | DINO | MAE | MoCo-v3 | MSN |
| 0 | 73.2 | 73.4 | 73.7 | 71.3 | 73.7 | 74.0 | 61.2 | 71.1 |
| | 70.3 | 72.3 | 54.2 | 67.6 | 40.7 | 45.9 | 46.0 | 48.4 |
| | 74.1 | 73.0 | 74.5 | 73.3 | 73.2 | 73.5 | 57.0 | 74.2 |
| 1 | 76.0 | 75.2 | 77.2 | 76.3 | 75.2 | 75.0 | 76.1 | 75.9 |
| | 75.7 | 74.8 | 76.6 | 76.2 | 75.3 | 54.0 | 75.2 | 76.8 |
| | 75.8 | 75.1 | 77.0 | 76.1 | 77.3 | 75.4 | 76.2 | 77.0 |
| 2 | 76.5 | 76.3 | 75.7 | 76.5 | 75.7 | 77.4 | 73.1 | 75.4 |
| | 76.8 | 76.4 | 76.6 | 77.0 | 76.3 | 66.3 | 72.9 | 76.3 |
| | 76.3 | 74.8 | 78.1 | 76.9 | 78.0 | 77.9 | 77.4 | 77.6 |
| 3 | 75.7 | 76.3 | 76.5 | 76.1 | 75.7 | 78.4 | 74.8 | 76.0 |
| | 76.8 | 76.4 | 77.2 | 77.1 | 76.2 | 72.5 | 75.1 | 76.5 |
| | 76.3 | 75.8 | 77.1 | 77.8 | 78.0 | 78.7 | 77.2 | 78.8 |
| 4 | 76.0 | 75.9 | 76.7 | 76.5 | 77.0 | 75.2 | 76.1 | 77.7 |
| | 76.9 | 76.7 | 76.8 | 77.5 | 77.2 | 75.6 | 75.8 | 78.0 |
| | 76.9 | 76.7 | 78.0 | 78.1 | 79.3 | 78.7 | 77.5 | 79.1 |
| 5 | 76.6 | 76.5 | 75.0 | 77.6 | 77.1 | 78.8 | 76.5 | 78.0 |
| | 77.3 | 77.1 | 76.0 | 78.6 | 78.0 | 78.8 | 76.6 | 78.4 |
| | 77.7 | 77.5 | 78.1 | 79.3 | 79.1 | 77.7 | 78.4 | 79.8 |
| 6 | 76.7 | 76.7 | 77.6 | 78.1 | 78.2 | 74.6 | 77.8 | 78.5 |
| | 77.1 | 77.0 | 78.1 | 78.5 | 79.3 | 74.9 | 78.1 | 79.3 |
| | 78.2 | 78.0 | 78.6 | 79.7 | 79.3 | 79.4 | 79.0 | 79.6 |
| 7 | 77.6 | 77.5 | 78.7 | 79.0 | 79.0 | 76.1 | 78.0 | 79.9 |
| | 78.0 | 77.9 | 78.8 | 79.8 | 80.0 | 77.8 | 78.7 | 80.0 |
| | 78.5 | 78.4 | 79.3 | 80.0 | 79.6 | 78.4 | 79.2 | 80.0 |
| 8 | 78.8 | 78.6 | 79.0 | 80.2 | 80.4 | 78.8 | 79.1 | 80.5 |
| | 78.9 | 78.6 | 79.0 | 80.6 | 80.8 | 80.0 | 79.8 | 80.7 |
| | 79.2 | 79.0 | 79.0 | 80.5 | 79.9 | 79.6 | 79.8 | 79.8 |
| 9 | 79.3 | 78.9 | 79.5 | 81.0 | 80.8 | 80.0 | 80.1 | 80.7 |
| | 79.3 | 78.9 | 79.3 | 81.1 | 81.1 | 80.1 | 80.7 | 81.0 |
| | 79.1 | 79.1 | 78.8 | 80.8 | 80.0 | 80.3 | 79.9 | 79.9 |
| 10 | 79.9 | 79.4 | 79.8 | 81.4 | 80.9 | 79.9 | 80.8 | 80.6 |
| | 79.5 | 78.9 | 79.4 | 81.5 | 81.3 | 80.3 | 81.2 | 80.9 |
| | 79.3 | 79.2 | 78.5 | 81.0 | 80.0 | 79.8 | 80.3 | 79.8 |
| 11 | 79.2 | 78.9 | 80.4 | 81.4 | 80.8 | 80.3 | 80.8 | 81.0 |
| | 78.6 | 78.2 | 80.3 | 81.3 | 81.3 | 80.2 | 80.7 | 81.2 |
| | 78.8 | 78.6 | 78.9 | 81.3 | 80.1 | 79.8 | 80.4 | 80.0 |

Table 1 shows results for ViT-B/16 architectures, across all layers, and for all features: *keys* ($k$), *queries* ($q$), and *values* ($v$). We highlight, for a given pre-training, in green best results, in blue the following five best results, and in red the three worst ones. Overall, results are very similar across all pre-trainings.

Although the variation is very small, each pre-training manages to beat vanilla ViTs. Besides, *keys* from the 11th layer are not necessarily the best representations. Even if they achieve very good results, other ones such as *queries* from the 10th layer are on par.

## B GRAPH ANALYSIS

In this section we study the performance impact when we use different forms of graphs. We will test three different forms of graphs: RBF, KNN, and similarity (see Section 3). We will also add color information to the graph, as in Melas-Kyriazi et al. (2022)

$$A = A_{feat} + \lambda A_{color}$$

where $A_{feat}$ is the adjacency matrix for the graph defined using ViT features, $\lambda$ is a hyperparameter, and $A_{color}$ is a KNN graph defined with color information. The similarity graph has a hyperparameter $\tau$ that we tune on VOC07, and find that $\tau = 0.8$ achieves best results (see Figure 8). Hence, we will use this value for the $\lambda$ analysis. We find that mixing the features graph with the color graph, as in Melas-Kyriazi et al. (2022), only decreased the performance of the algorithm. Thus, we conducted all our analysis using the default RBF graph.

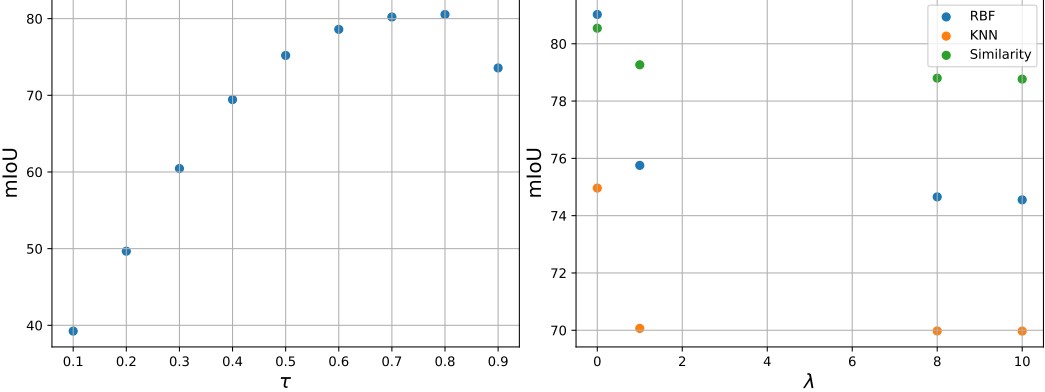

Figure 8: (left) $\tau$ analysis for similarity graph. (right) $\lambda$ analysis on VOC07.

## C RESULTS PER CATEGORY

Table 2 and 3 shows mIoU results per category for COCO-panoptic. As expected, categories that have complex masks, usually with high convex hull over area ratio, achieve the lowest scores, i.e. bicycle. Also, very small objects, that have small ends with sizes below a patch area perform very bad, i.e. toothbrush, fork. On the other hand, objects with a low convex hull over area ratio achieve better results, i.e. bus, train. Usually, these objects are alone on the image and located at the center with a clear background. The performance of the algorithm is also related to the nature of ImageNet images.

We also analyse the statistics of the user interaction per category. Let $I$ be the user interaction over the whole dataset. We report the mean $m(I)$, and standard deviation $\sigma(I)$ percentage of labeled patches by the user. We denote by $I^+$, and $I^-$ the correctly, and incorrectly labeled patches respectively. Finally, we denote by Tot, the total number of patches per category. We notice that the category with the highest number of incorrectly labeled patches is `Fork`, which also has one of the lowest average number of patches, and achieves one of the lowest mIoU. This can be explained as fork labels are usually very thin, and do not cover entire patches.

Table 2: Pascal VOC07 results, and number of interactions per category. Let $I$ be the user interaction over the whole dataset. We define $m(I)$ as the mean percentage of $I$, and $\sigma(I)$ as its standard deviation. We denote by $m(I^+)$ ($m(I^-)$) the mean over the correctly (incorrectly) labeled interactions. The mean, and standard deviation, of the total number of patches are respectively denoted by m(Tot), $\sigma$(Tot). We use patches of size $16 \times 16$.

| Category | mIoU | $m(I^+)$ | $m(I^-)$ | $\sigma(I)$ | $m$(Tot) | $\sigma$(Tot) |
|---|---|---|---|---|---|---|
| Background | 90.3 | 7% | 0.0% | 3% | 3001 | 1182 |
| person | 66.4 | 19% | 0.6% | 10% | 289 | 432 |
| bicycle | 58.2 | 19% | 2.1% | 12% | 341 | 684 |
| car | 62.3 | 19% | 0.3% | 8% | 177 | 348 |
| motorcycle | 71.1 | 15% | 0.3% | 8% | 516 | 573 |
| airplane | 74.7 | 21% | 1.8% | 16% | 316 | 312 |
| bus | 77.2 | 16% | 0.4% | 11% | 744 | 786 |
| train | 79.3 | 12% | 0.1% | 6% | 740 | 718 |
| truck | 70.3 | 16% | 0.3% | 8% | 405 | 609 |
| boat | 61.8 | 20% | 0.6% | 11% | 233 | 271 |
| traffic light | 56.8 | 22% | 0.3% | 9% | 116 | 199 |
| fire hydrant | 83.5 | 15% | 0.0% | 7% | 522 | 453 |
| stop sign | 88.8 | 13% | 0.0% | 9% | 639 | 638 |
| parking meter | 81.0 | 14% | 0.2% | 8% | 381 | 328 |
| bench | 64.3 | 17% | 0.8% | 14% | 428 | 632 |
| bird | 67.8 | 21% | 0.5% | 12% | 177 | 247 |
| cat | 84.4 | 12% | 0.0% | 7% | 881 | 794 |
| dog | 80.3 | 14% | 0.1% | 8% | 649 | 633 |
| horse | 65.6 | 18% | 0.4% | 7% | 289 | 320 |
| sheep | 64.4 | 18% | 0.5% | 8% | 184 | 286 |
| cow | 68.0 | 18% | 0.3% | 8% | 232 | 255 |
| elephant | 70.2 | 16% | 0.0% | 8% | 471 | 572 |
| bear | 84.0 | 12% | 0.0% | 7% | 1163 | 1219 |
| zebra | 62.9 | 17% | 0.2% | 7% | 394 | 522 |
| giraffe | 67.7 | 23% | 1.1% | 10% | 280 | 321 |
| backpack | 59.7 | 21% | 1.9% | 15% | 116 | 184 |
| umbrella | 75.0 | 18% | 0.5% | 9% | 302 | 466 |
| handbag | 65.1 | 19% | 1.1% | 11% | 75 | 53 |
| tie | 71.4 | 20% | 1.1% | 15% | 171 | 124 |
| suitcase | 72.5 | 16% | 0.2% | 8% | 257 | 390 |
| frisbee | 87.1 | 27% | 0.0% | 16% | 81 | 73 |
| skis | 37.7 | 33% | 9.9% | 18% | 53 | 40 |
| snowboard | 69.5 | 24% | 1.4% | 10% | 176 | 190 |
| sports ball | 75.2 | 17% | 0.0% | 12% | 64 | 47 |
| kite | 67.7 | 21% | 1.0% | 10% | 96 | 71 |
| baseball bat | 58.4 | 30% | 3.6% | 10% | 76 | 59 |
| baseball glove | 74.3 | 18% | 0.1% | 8% | 136 | 299 |
| skateboard | 62.9 | 26% | 4.2% | 16% | 153 | 292 |
| surfboard | 67.5 | 23% | 1.7% | 17% | 150 | 267 |
| tennis racket | 71.9 | 22% | 0.6% | 10% | 101 | 115 |
| bottle | 59.2 | 22% | 0.6% | 10% | 117 | 162 |

Table 3: Same as Table 2 for remaining categories.

| Category | mIoU | $m(\boldsymbol{I}^+)$ | $m(\boldsymbol{I}^-)$ | $\sigma(\boldsymbol{I})$ | $m(\text{Tot})$ | $\sigma(\text{Tot})$ |
|---|---|---|---|---|---|---|
| wine glass | 59.6 | 17% | 0.2% | 7% | 105 | 83 |
| cup | 68.8 | 18% | 0.4% | 12% | 108 | 99 |
| fork | 48.4 | 26% | 5.2% | 15% | 104 | 129 |
| knife | 51.8 | 33% | 4.1% | 17% | 80 | 83 |
| spoon | 52.8 | 26% | 3.3% | 15% | 71 | 59 |
| bowl | 74.4 | 16% | 0.3% | 9% | 583 | 849 |
| banana | 62.0 | 16% | 0.1% | 8% | 234 | 412 |
| apple | 63.7 | 15% | 0.0% | 7% | 194 | 221 |
| sandwich | 75.2 | 11% | 0.0% | 7% | 364 | 382 |
| orange | 65.4 | 14% | 0.0% | 8% | 189 | 312 |
| broccoli | 58.5 | 18% | 0.0% | 7% | 139 | 208 |
| carrot | 61.6 | 23% | 0.1% | 9% | 140 | 200 |
| hot dog | 69.6 | 18% | 0.1% | 11% | 353 | 470 |
| pizza | 80.4 | 13% | 0.0% | 8% | 789 | 965 |
| donut | 56.8 | 23% | 0.3% | 10% | 68 | 99 |
| cake | 74.3 | 16% | 0.1% | 8% | 411 | 539 |
| chair | 55.3 | 20% | 1.2% | 14% | 125 | 209 |
| couch | 77.1 | 14% | 0.1% | 7% | 419 | 526 |
| potted plant | 59.2 | 18% | 0.7% | 8% | 121 | 181 |
| bed | 78.8 | 9% | 0.4% | 5% | 1137 | 853 |
| dining table | 72.7 | 13% | 0.2% | 8% | 868 | 985 |
| toilet | 70.2 | 15% | 0.2% | 6% | 329 | 346 |
| tv | 76.8 | 15% | 0.1% | 8% | 499 | 660 |
| laptop | 70.6 | 14% | 0.4% | 9% | 340 | 378 |
| mouse | 65.5 | 15% | 0.0% | 8% | 79 | 67 |
| remote | 59.4 | 20% | 1.6% | 14% | 78 | 65 |
| keyboard | 75.3 | 15% | 0.0% | 7% | 277 | 353 |
| cell phone | 72.1 | 17% | 0.2% | 12% | 341 | 538 |
| microwave | 72.8 | 19% | 0.0% | 6% | 127 | 130 |
| oven | 69.2 | 17% | 0.7% | 10% | 290 | 311 |
| toaster | 65.0 | 10% | 0.0% | 6% | 644 | 592 |
| sink | 69.5 | 18% | 0.3% | 9% | 198 | 244 |
| refrigerator | 75.6 | 12% | 0.1% | 6% | 645 | 914 |
| book | 60.7 | 19% | 0.3% | 9% | 116 | 148 |
| clock | 66.7 | 18% | 1.2% | 13% | 157 | 215 |
| vase | 60.9 | 19% | 0.0% | 11% | 201 | 312 |
| scissors | 56.7 | 15% | 0.1% | 10% | 666 | 869 |
| teddy bear | 78.3 | 14% | 0.0% | 8% | 537 | 628 |
| hair drier | 71.0 | 19% | 0.0% | 8% | 200 | 159 |
| toothbrush | 47.9 | 23% | 1.2% | 9% | 48 | 6 |

## D   ADDITIONAL RESULTS

In this section we present additional results for COCO-panoptic. Figure 9 shows results when using only a fraction of labels. Figure 10 shows results when we add noise to object interactions. Figure 11 shows additional results using one patch only. Figure 12 shows failure cases. Figure 13 to 16 show additional random results for the default configuration.

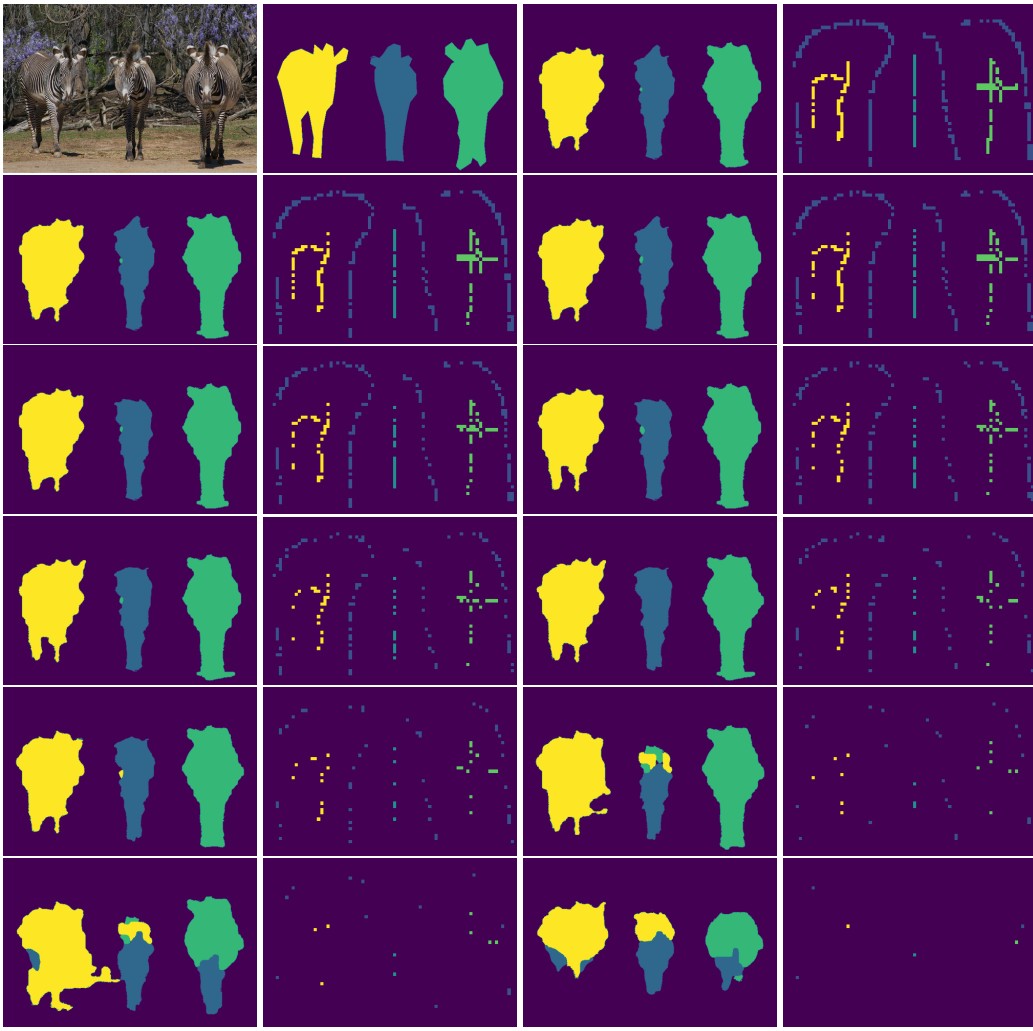

Figure 9: Analysis of SALT robustness against using a fraction of the handmade labels using MAE-B/16 keys from 11th layer. From left to right, original COCO-panoptic image (000000011760) and ground truth labels, the predicted labels and interactions for $p \in [0.1, 0.99]$.

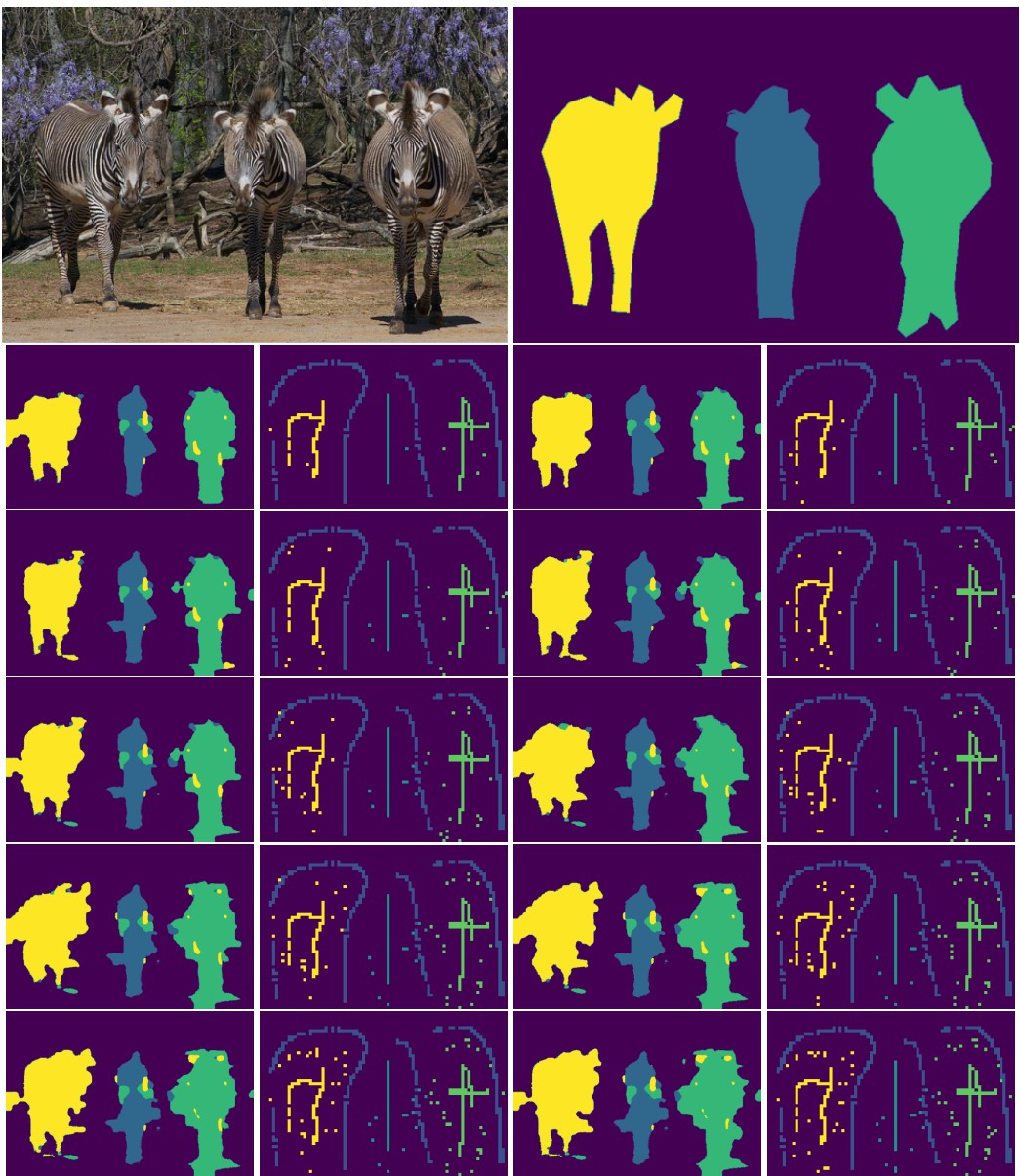

Figure 10: Analysis of SALT robustness against noise. From left to right, original COCO-panoptic image (000000011760) and ground truth labels, the predicted labels and interactions for $p \in [0.05, 0.5]$.

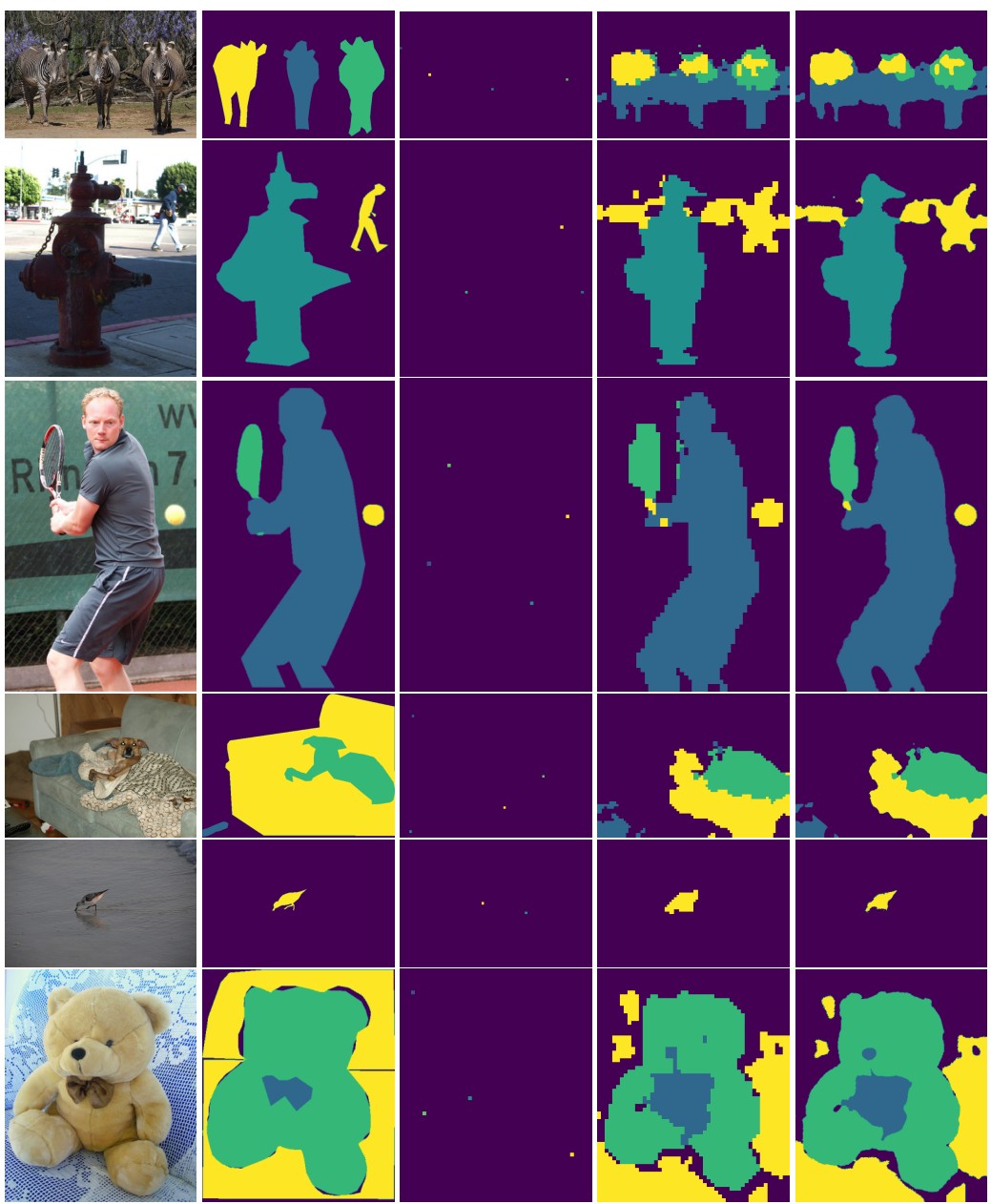

Figure 11: Same as Figure 7.

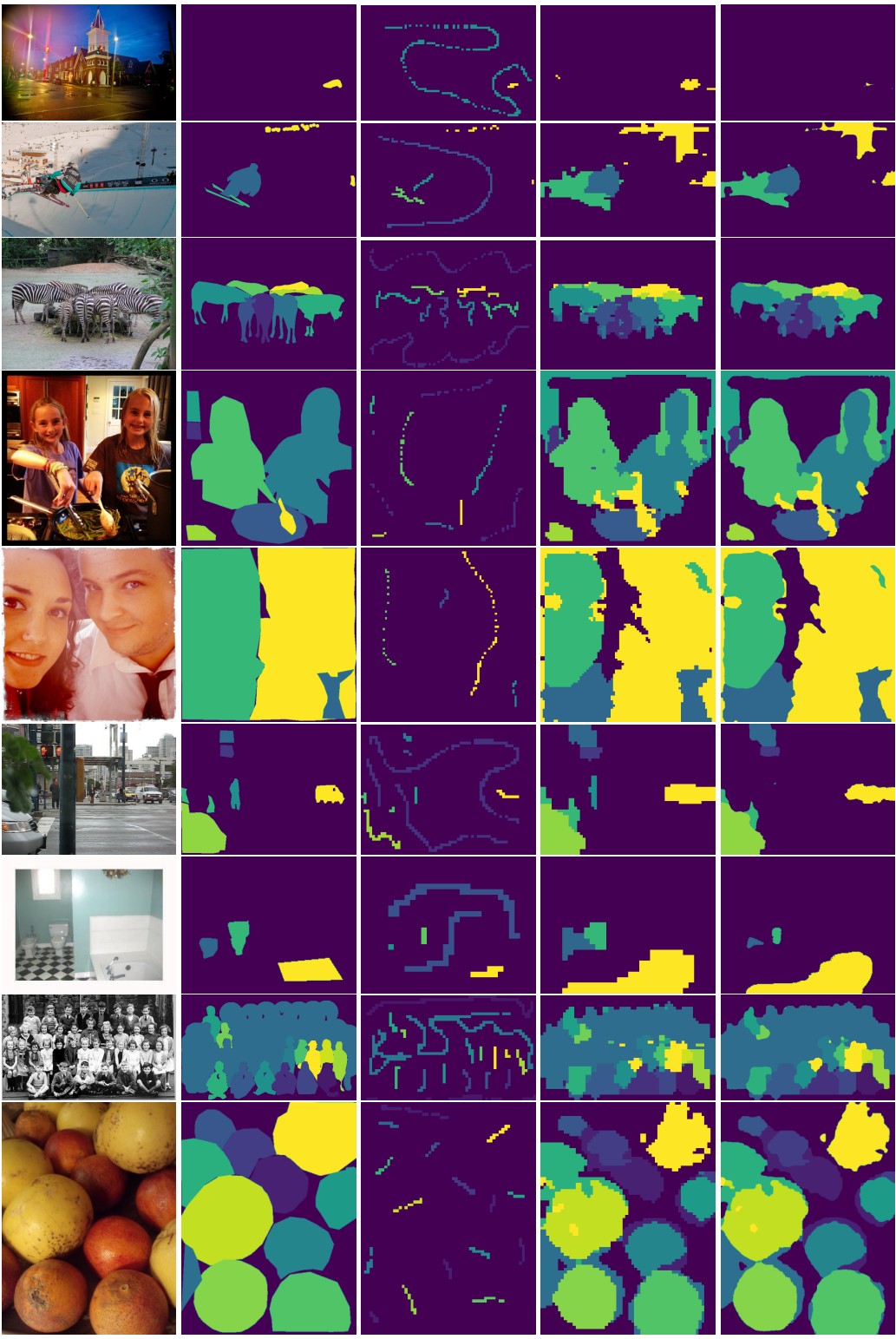

Figure 12: Failure cases for COCO-panoptic. Same as Figure 3.

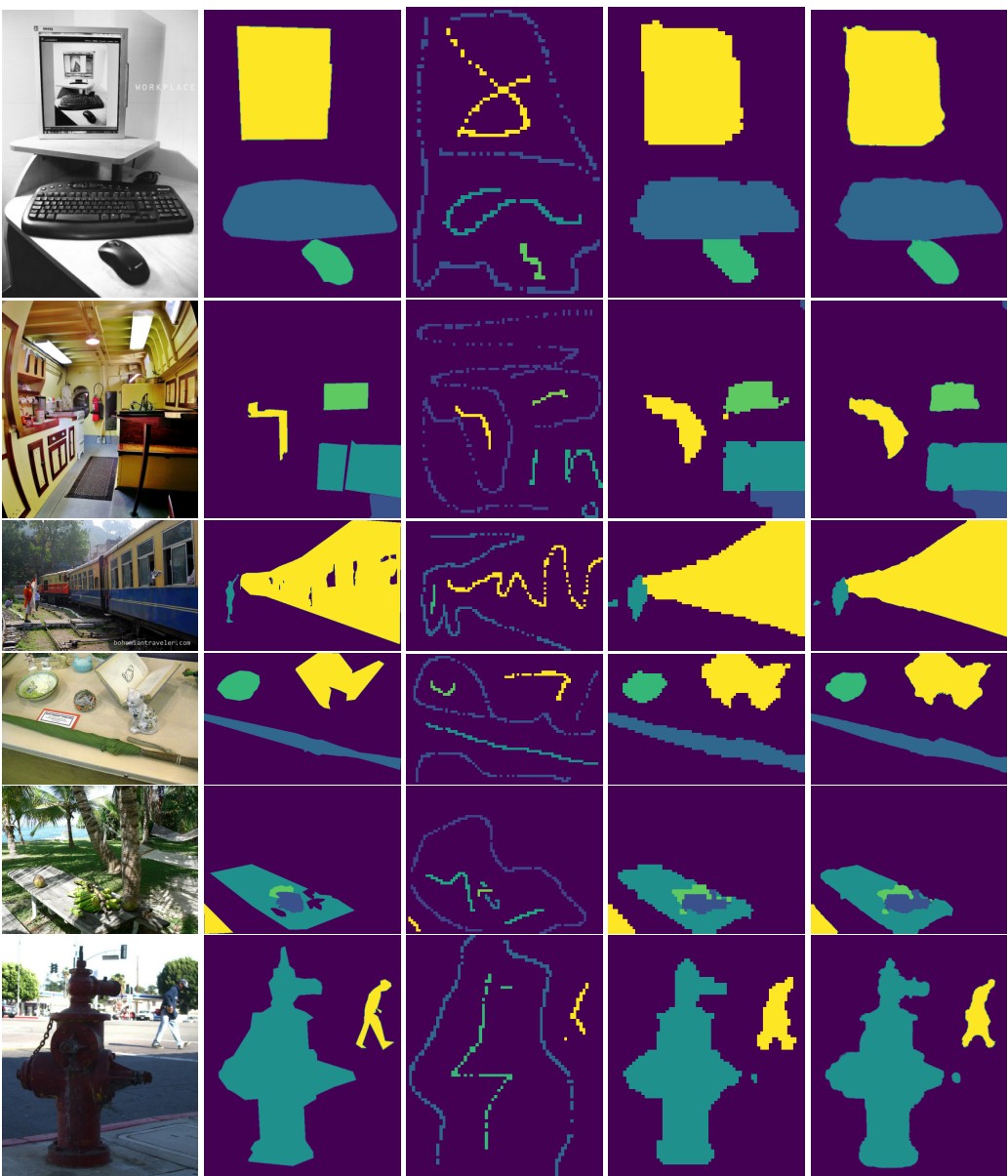

Figure 13: Same as Figure 3.

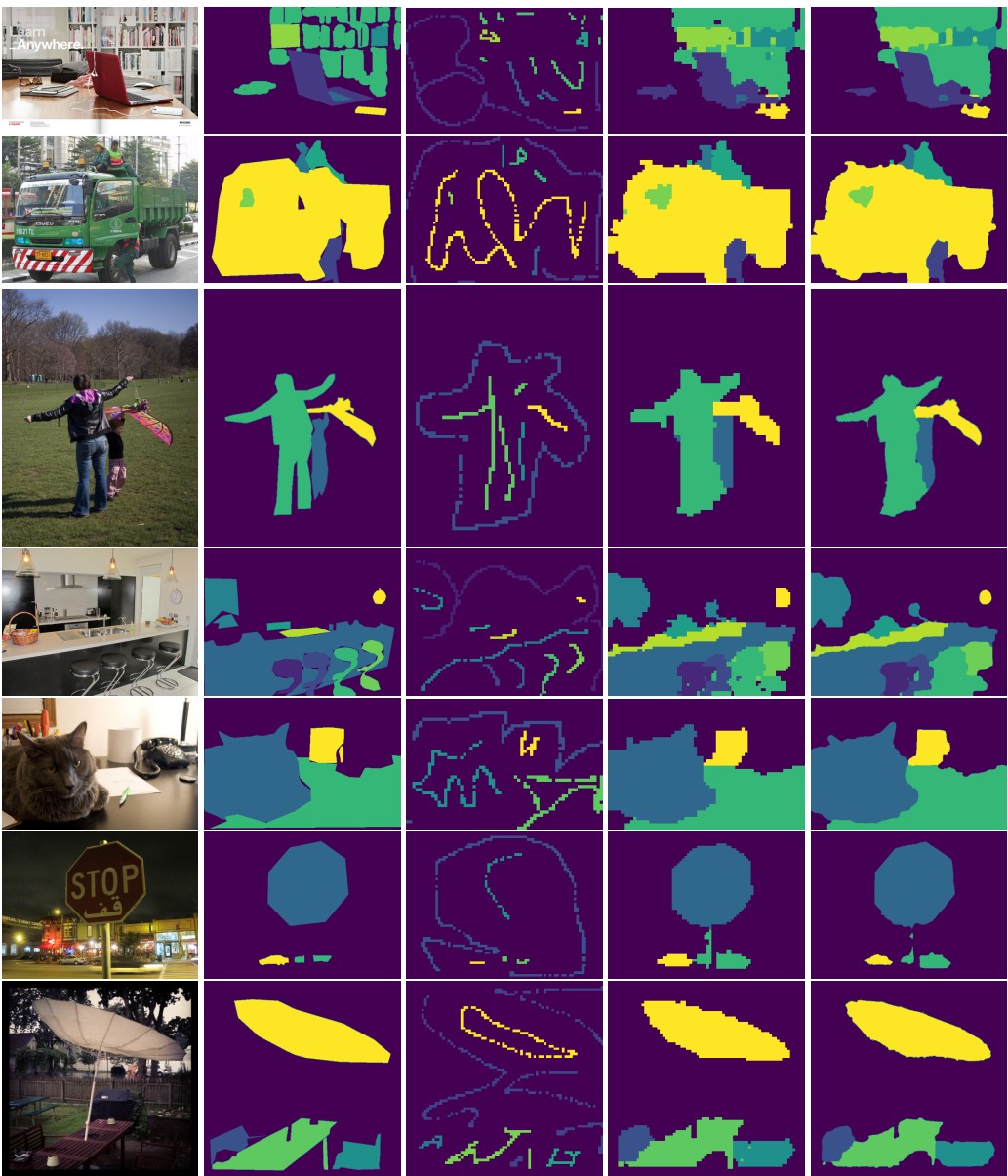

Figure 14: Same as Figure 3.

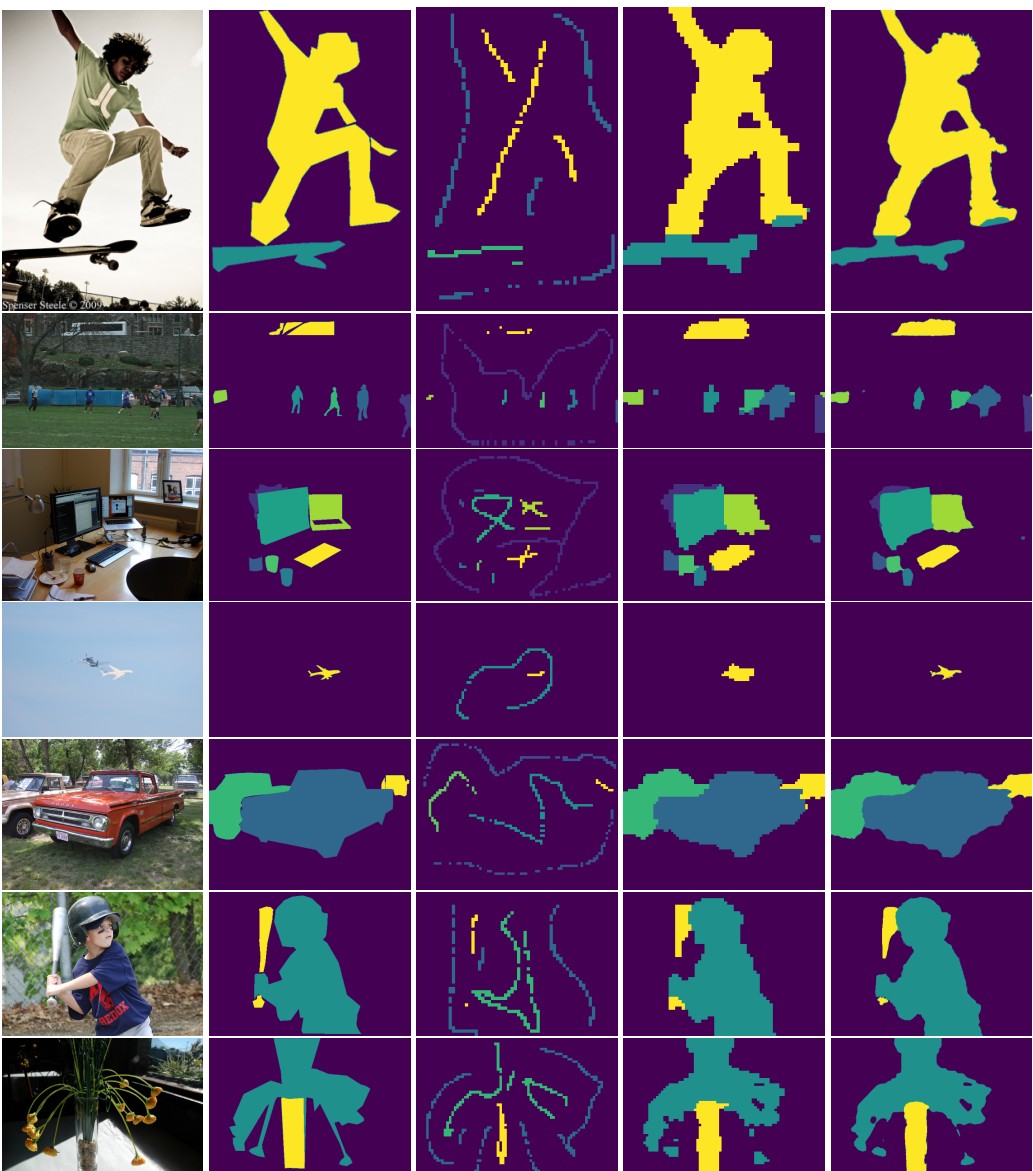

Figure 15: Same as Figure 3.

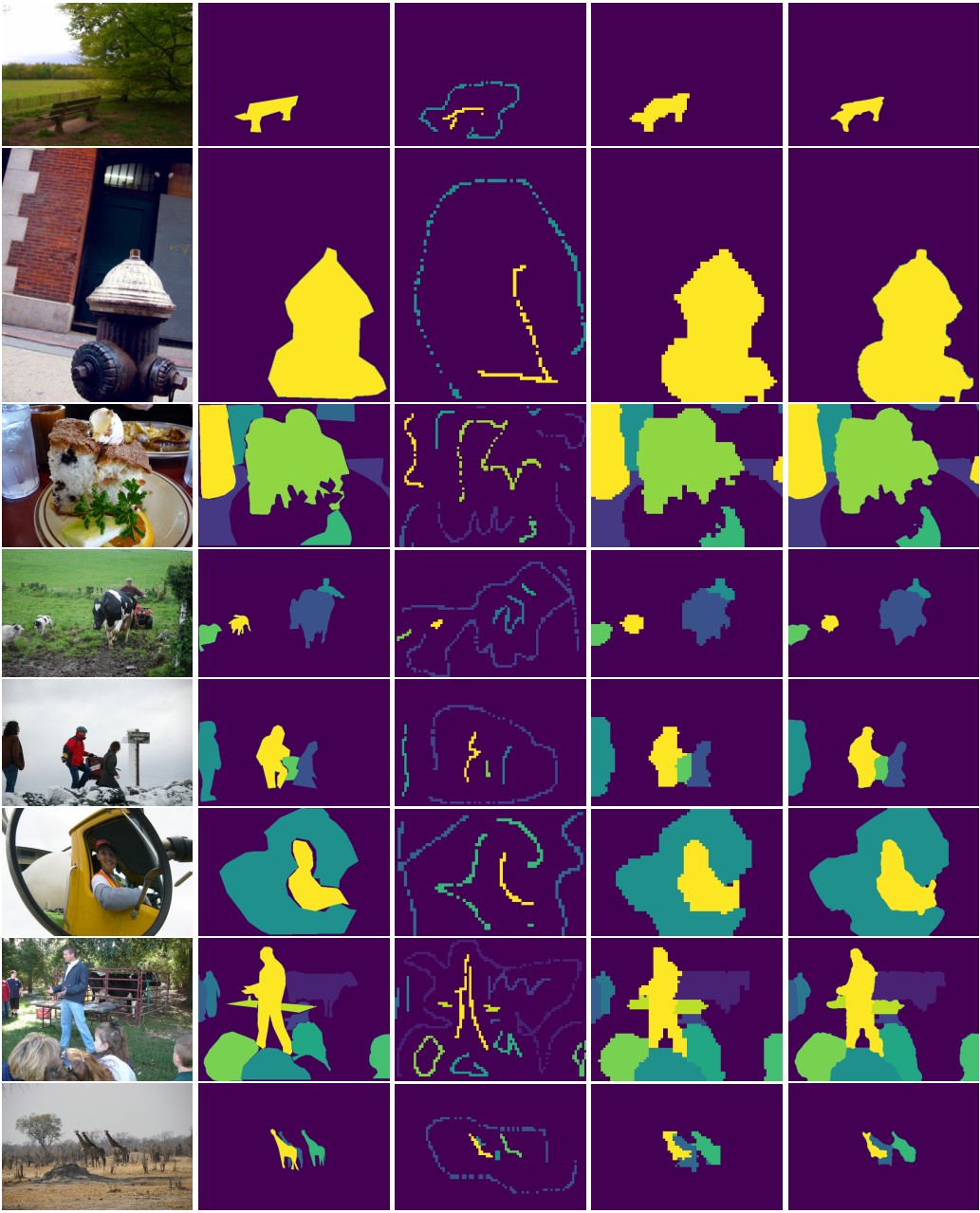

Figure 16: Same as Figure 3.

