# OpenReview forum: "Multi-instance Interactive Segmentation with Self-Supervised Transformer"
_ICLR.cc/2023/Conference — Submitted to ICLR 2023_

### Official Review · Reviewer_KmRX · 2022-10-23

**Confidence:** 4
**Correctness:** 3
**Technical Novelty And Significance:** 2
**Empirical Novelty And Significance:** 2
**Recommendation:** 3

**Clarity, Quality, Novelty And Reproducibility:**

This paper describes clearly the insight and how their proposed algorithm works. They also analyze their method in complex situations including adding noise and removing a fraction of annotations. However, as I mentioned above, this work is a combination of available modules thus not being considered an enormous novelty.

**Strength And Weaknesses:**

The main contribution of this work is applying feature representation of self-supervised ViTs and Label Propagation for interactive segmentation tasks. The authors also assess the performance of their algorithm in extreme circumstances with very sparse information on the objects’ position. In addition, the methodology of annotating datasets in an interactive way is carefully described.

This paper is presented in a clear, coherent manner.  Each module is explained thoroughly and experiments provide detailed information.

However, the contributions are not much impressive and not considerable as this paper is just a combination of previous modules. For instance, the authors utilize the last ViT-B layer for building a graph and also follow the prior work (Simeoni et al. 2021).  Moreover, the authors reuse the Label Propagation without significant modification and contribution.

Experiments lack comparison with other methods in interactive object segmentation. For instance,  f-BRS: Rethinking Backpropagating Refinement for Interactive Segmentation - CVPR 2020 or EdgeFlow: Achieving Practical Interactive Segmentation with Edge-Guided Flow - ICCV 2021. It is difficult to convince that this proposal stands out as a state-of-the-art method.


**Summary Of The Paper:**

In this paper, the authors try to address the multi-instance interactive segmentation task instead of salient interactive segmentation via the Label Propagation algorithm with patch feature representation from Transformers.

In particular, the authors utilize keys from the last ViT-B layer for feature representation which is later leveraged to construct a graph. A user is assigned to draw scribbles for different objects in an image and apply the Label Propagation algorithm to propagate information through the graph and the remaining nodes.
The main contribution of this paper is that this work evaluates the representation power of self-supervised ViTs for interactive segmentation tasks.

Experiments illustrate the effectiveness of the last ViT-B layer with different pre-trainings and demonstrate the robustness against noise and using different ratios of interactive annotations.


**Summary Of The Review:**

The main contribution of this work is utilizing feature representation of ViTs and Label Propagation for interactive segmentation tasks. Although experiments comprehensively analyze the effectiveness of different pre-trainings as well as the performance of proposed methods in extreme conditions, the contribution of this work is minor and does not impress me a lot. Moreover, experiments do not compare with current methods in interactive segmentation tasks which can not prove outstanding results.

---

### Official Review · Reviewer_JhQx · 2022-10-24

**Confidence:** 4
**Correctness:** 3
**Technical Novelty And Significance:** 2
**Empirical Novelty And Significance:** 2
**Recommendation:** 3

**Clarity, Quality, Novelty And Reproducibility:**

- The clarity of the work is fair. But it is not clear whether the scribbles need to cover all semantic classes in an image for the algorithm to work.
- The novelty of the work is not very high.
- The work can be reproduced by the description provided.
- The overall writing still has room for improvement.

**Strength And Weaknesses:**

Strength:
- The paper shows that pre-trained vision features can be used for semi-supervised multi-class/multi-instance segmentation, where the supervision signals can be as simple as scribbles or even sparse points, which are obtained with low manual cost.
- The work investigated multiple choices in the system, including the pre-train method to obtain vision features (both supervised and self-supervised), the metrics to build the graph (similarity, RBF, and KNN), and the quality of the supervision signal (partial missing and noisy).

Weakness:
- The novelty of the work is not high, as it is a simple extension to existing methods (unsupervised segmentation using pre-trained vision features), and there is no big challenge in this extension. The work follows previous work in graph construction and label propagation.
- The work is not compared with other work. One reason might be that the problem setting is too specific. But how does this setting compare with other interactive segmentation settings?
- The authors simply ignore small objects, but the segmentation accuracy of small objects is important to the segmentation problem.


**Summary Of The Paper:**

This paper presents an interactive algorithm for multi-class/multi-instance segmentation based on the vision features obtained from self-supervised pre-training. Previously, such vision features, e.g. DINO features, have been used for unsupervised segmentation. This work extends previous work to the interactive segmentation of multiple instances. Similar to prior work, a graph is constructed based on the features learned in the pre-training stage. Then, class-differentiated scribbles from users are used to initialize partial labels. Finally, a label propagation algorithm (Zhou et al., 2004) is adopted to propagate partial labels to the entire image.  Experiments on VOC 2007 and COCO-panoptic show that the proposed method can achieve reasonable performance and the method is quite robust against noise.

**Summary Of The Review:**

This paper presents a new way to do interactive multi-instance segmentation. However, considering the prior arts on DINO-based unsupervised segmentation and interactive segmentation methods, and the fact that the key components including graph construction and label propagation are not new, the novelty of the work is not high.

---

### Official Review · Reviewer_Uyni · 2022-10-25

**Confidence:** 3
**Correctness:** 3
**Technical Novelty And Significance:** 2
**Empirical Novelty And Significance:** 2
**Recommendation:** 6

**Clarity, Quality, Novelty And Reproducibility:**

Strength
 + Overall clear presentation in problem statement, proposed method, and experiment results

Weaknesses
 + reproducibility is poor without the release of code by the author

**Strength And Weaknesses:**

Strength
 + An interesting application of ViT in semi-supervised interactive segmentation framework
 + Comprehensive hyper-parameter tuning and experimentation with extreme settings

Weaknesses
 + Although already stated in the paper with understandable reasons, it's still worth pointing out that this work is lacking comparison with existing works and alternative solutions.
 + No evidence is provided as far as I know to substantiate the claim that "representation of ViT are not able to segment images, with several classes and object instance, in an unsupervised fashion yet". I believe the claim is true, but proof is needed in the paper.

**Summary Of The Paper:**

This work proposes an interactive segmentation framework that leverages a pre-trained ViT model to perform semi-supervised segmentation tasks on otherwise challenging images. Due to the nature of the proposed framework and ambiguity in evaluation for interactive segmentation framework, no baseline is provided or compared against. The author provides some ablation study within the proposed framework.

**Summary Of The Review:**

It is an interesting application of ViT in semi-supervised interactive segmentation framework. Comprehensive hyper-parameter tuning and experimentation with extreme settings are provided within the proposed framework. To facilitate reproducibility, I strongly recommend the author publish their code upon acceptance.

---

### Official Review · Reviewer_yrjR · 2022-10-25

**Confidence:** 4
**Clarity, Quality, Novelty And Reproducibility:** The quality, clarity, and originality…
**Correctness:** 2
**Technical Novelty And Significance:** 1
**Empirical Novelty And Significance:** 2
**Recommendation:** 3

**Strength And Weaknesses:**

Strength

1.	Multi-instance segmentation with less label and human interaction is interesting and practically useful research topic.

Weaknesses

1.	The paper is not organized and written well so that it is hard to follow and understand the idea.
2.	The motivation is clearly introduced, thus the underlying principle of proposing the SALT is not understandable.
3.	The experiments are not convincing, no comparison with other state-of-the-art methods.


**Summary Of The Paper:**

The paper proposes a semi-supervised segmentation with self-supervised attention layers for interactive multi-instance segmentation. Remarkable experiment results are obtained when evaluating on Pascal and COCO-panoptic data with very noisy and sparse labels.

**Summary Of The Review:**

Based on the weaknesses mentioned above, my current decision is reject.

---

### Decision · Program_Chairs · 2023-01-20

**Decision:**

Reject

**Justification For Why Not Higher Score:**

Three reviewers give reject rating and one gives borderline rating. They express the similar concerns on the novelty, lack of comparison with existing works and writing quality. The authors do not provide any response to these questions.

**Justification For Why Not Lower Score:**

N/A

**Metareview: Summary, Strengths And Weaknesses:**

Summary:

This paper proposes to leverage the self-attention maps of self-supervised trained ViT models to generate segmentation masks for images with multiple objects. This is inspired by the findings from existing works (e.g., DINO) that: the self-attention maps of the self-supervised ViTs can highlight the salient objects. However, such attention maps cannot distinguish multiple objects within the same image and thus are not ready to utilize for segmenting images with multiple objects. To relieve such a difficult, this paper proposes to introduce sparse human labels and transforms the problem to a weakly-supervised learning one. The proposed method takes the attention map as seeds and applies label propagation methods to generate the segmentation maps.

Strengths:

- The idea is clean and simple.
- The experiment results are encouraging.

Weakness:

- The novelty is limited. The label propagation method has been extensively studied in the existing works on weakly-supervised image segmentation.
- Lack of comparison with SOTA. Thus it is hard to benchmark the performance of the proposed method.
- The writing quality is not good.